# *Houttuynia cordata* Thunb. Extracts Alleviate Atherosclerosis and Modulate Gut Microbiota in Male Hypercholesterolemic Hamsters

**DOI:** 10.3390/nu16193290

**Published:** 2024-09-28

**Authors:** Yuhong Lin, Chufeng He, Jianhui Liu, Hau-Yin Chung, Zhen-Yu Chen, Wing-Tak Wong

**Affiliations:** 1School of Life Sciences, The Chinese University of Hong Kong, Hong Kong, China; 1155099814@link.cuhk.edu.hk (Y.L.); 1155134757@link.cuhk.edu.hk (C.H.); anthonychung@cuhk.edu.hk (H.-Y.C.); zhenyuchen@cuhk.edu.hk (Z.-Y.C.); 2State Key Laboratory of Agrobiotechnology, The Chinese University of Hong Kong, Hong Kong, China; 3Collaborative Innovation Center for Modern Grain Circulation and Safety, Jiangsu Province Engineering Research Center of Edible Fungus Preservation and Intensive Processing, College of Food Science and Engineering, Nanjing University of Finance and Economics, Nanjing 210023, China; 9120191155@nufe.edu.cn; 4Shenzhen Research Institute, The Chinese University of Hong Kong, Shenzhen 518172, China

**Keywords:** dietary supplement, vascular diseases, gut microbiota, hypercholesteremia

## Abstract

**Background and Aims:** Hypercholesterolemia leads to cardiovascular diseases and atherosclerosis. Previous studies have highlighted the crucial role of gut microbiota in alleviating atherosclerosis progression and reducing plasma cholesterol. However, the protective effects of *Houttuynia cordata* Thunb (HCT), a well-known fishy Chinese herb, against hypercholesterolemia and vasculopathy remain largely unknown. This study aims to explore the effects of HCT extracts on vascular health and gut microbiota in golden Syrian hamsters with hypercholesterolemia. **Methods:** The hypercholesterolemia hamster model was established by feeding with a high-cholesterol diet. Aqueous or ethanolic HCT extracts were mixed with diet and concurrently given to hamsters for Six weeks. Plasma lipid profiles were evaluated. Aortas were collected to detect fatty streak areas. Feces were collected to analyze the abundance of microorganisms in the gut microbiota. **Results:** HCT ethanolic extract treatment remarkedly decreased plasma levels of total cholesterol and high-density lipoprotein cholesterol in hypercholesterolemic hamsters. Notably, both aqueous and ethanolic extracts of HCT reduced atherosclerotic plaques in hamsters fed with a high-cholesterol diet. Strikingly, the effects of HCT ethanolic extract in reducing atherosclerotic plaques are greater than aqueous extract. Furthermore, at the phylum level, the relative abundance of *Firmicutes* was decreased in hamsters treated with aqueous and ethanolic extracts of HCT. By contrast, the abundance of *Bacteroidetes* was increased by HCT treatment. At the family level, HCT extract favourably modulated the relative abundance of *Porphyromonadaceae* and *Bacteroidales_S24-7_group*. These findings indicate that HCT extracts may facilitate the growth of short-chain fatty acids-producing bacteria to alter gut microbiota composition, contributing to the reduction of plasma lipid levels. **Conclusions:** This study offers evidence demonstrating the effects of HCT extracts on alleviating atherosclerosis and lowering plasma cholesterol levels in the male hypercholesterolemic hamster model, offering novel insights into the pharmacological effects and promoting the application of HCT. This study highlights the potential of HCT as a dietary supplement to alleviate atherosclerosis, lower plasma cholesterol, and modulate the abundance of microorganisms in gut microbiota.

## 1. Introduction

The events of cardiovascular deaths have been increasing in recent years worldwide [1]. Hypercholesterolemia is characterized by elevated concentrations of cholesterol, such as non-high-density lipoprotein cholesterol (non-HDL-C), in the blood [2]. Increasing plasma total cholesterol levels contribute significantly to cardiovascular mortality [3]. Excessive cholesterol in the blood thickens the arterial wall and ultimately results in atherosclerosis, which is the most common vascular disease worldwide [4]. Notably, it has been reported that reducing plasma cholesterol profiles is an efficient therapeutic approach to reduce cardiovascular deaths, ultimately improving survival in patients suffering from vasculopathy [5,6].

Previous studies have reported that bacterial metabolites are tightly correlated with elevated total plasma cholesterol levels and atherosclerosis [7,8]. Accumulative findings have emphasized the critical role of gut microbiota in altering plasma lipid composition, particularly in the cholesterol level, through regulating bile acid metabolism [9]. Intriguingly, some recent studies emphasize the importance of modulating gut microbiota in preventing vascular diseases by regulating several metabolic and inflammatory processes [10]. Thus, it is a promising approach to improve vascular health via reversing gut microbiota disorders upon hypercholesterolemia stimulation.

Recently, functional foods and nutraceuticals have gained high popularity as non-pharmacological treatments to prevent hypercholesteremia and vascular diseases [11]. The negligible safety concern of bioactive compounds and traditional herbs prompts us to explore the potential of traditional herbal extracts to be developed as dietary supplements to alleviate hypercholesteremia and atherosclerosis. *Houttuynia cordata* Thunb. (HCT) is a food medicinal herb exhibiting various pharmacological properties such as anti-inflammatory [12], antioxidant [13], antitumor [14], and antiviral [15]. HCT is a distinguished traditional Chinese medicine listed in the Chinese Pharmacopoeia [16]. It is a perennial creeping herb with stoloniferous rhizomes, heart-shaped leaves, and a distinctive fishy odour, thriving in moist locations [17]. People in Guizhou, Sichuan Provinces in China favour its root as a cold dish, and people in Southeast Asia use its leaves to make soup. The HCT extracts contain phytochemicals such as essential oils, flavonoids, alkaloids, fatty acids, sterols, and polyphenolic acids [17]. The potential of HCT extracts to decrease blood glucose levels and lipid levels has been reported in previous studies. Furthermore, HCT has been reported to regulate immunity by improving immune barriers in different tissues. The potential of HCT in inhibiting tumour generation has also been found, while the underlying mechanism remains to be investigated [18]. Notably, one study has revealed that the ethanolic extracts of HCT significantly improve serum lipid profile and ameliorate lipid peroxidation in rats [19], indicating the beneficial effects of ethanolic extracts of HCT on improving lipid metabolism. Moreover, studies have illustrated that ethanolic extracts of HCT alleviate lipid accumulation in livers, emphasizing the therapeutic potential of HCT ethanolic extracts in attenuating the progression of fatty liver diseases [20]. By contrast, the effects of HCT on regulating cholesterol metabolism and alleviating atherosclerosis remain largely unknown. Hamsters have been proven to act in a more human-like pattern in the synthesis and excretion of cholesterol. However, few studies have reported the cholesterol-lowering effects of HCT extracts using hamster models.

Therefore, this study aimed to elucidate the effects of HCT aqueous extract and HCT ethanolic extract on lowering cholesterol and alleviating atherosclerosis using a hypercholesterolemia hamster model. Considering the crucial role of gut microbiota in ameliorating hypercholesterolemia-associated vascular diseases, this study also evaluated the effects of HCT extracts on modulating gut microbiota, particularly focusing on bacteria responsible for producing short-chain fatty acids (SCFA) [20]. This study provides valuable insights into the beneficial effects of HCT on alleviating atherosclerosis and hypercholesteremia.

Collectively, this current study presents the first line of evidence of the effects of HCT extracts on alleviating atherosclerosis and modulating cholesterol metabolism and gut microbiota in high-cholesterol diet-induced hypercholesterolemic hamsters. Briefly, HCT extracts markedly reduce the areas of atherosclerotic plaques in hypercholesterolemic hamsters, especially the ethanolic extracts of HCT. The total plasma cholesterol levels were significantly lowered after six weeks of HCT treatment. Encouragingly, the relative abundance of *Firmicutes* is decreased in HCT-treated hamsters, while the abundance of *Bacteroidetes* is increased by HCT treatment. Furthermore, HCT extracts favourably modulated the relative abundance of *Porphyromonadaceae* and *Bacteroidales_S24-7_group*, implicating that HCT supplementation might facilitate the growth of short-chain fatty acids-producing bacteria to prevent vascular damage caused by hypercholesteremia.

## 2. Materials and Methods

### 2.1. Houttuynia Cordata Thunb Extracts

The whole plant of *Houttuynia cordata Thunb,* including roots, stems, and leaves were used in the study. Food grade 10:1 aqueous extract (Batch No: XTY20210511) and 10:1 ethanolic extract (Batch No: XTY20210523) of *Houttuynia cordata* Thunb. were obtained commercially from Shaanxi Xintianyu Biotechnology Company Limited (Shaanxi, China). The extracts contain phytochemicals such as essential oils, flavonoids, alkaloids, fatty acids, sterols, and polyphenolic acids.

### 2.2. Animals

Protocols were approved by the University Animal Experimentation Ethics Committee (AEEC) of The Chinese University of Hong Kong (CUHK) under Ref No. 21-201-MIS. Forty-seven eight-week-old male Golden Syrian hamsters (*Mesocricetus auratus*; body weight 100–120 g) were divided into six groups (*n* = 7–9) at random. All hamsters were kept in an animal room maintained at 23 °C and a 12-h light and 12-h dark cycle with *ad libitum* access to food and water. Male hamsters were fed with a high-cholesterol diet to establish a hypercholesterolemia model. HCT extracts were directly mixed with the high-cholesterol diet and concurrently provided to the hamsters daily. Diet composition is shown in Table 1. The diet composition was determined according to the literature review [21,22]. Daily food consumption was monitored, and body weight was recorded twice a week. The fresh fecal samples for 16S rRNA gene sequencing analysis were collected from each hamster at the end of week 6. After overnight fasting, all hamsters were sacrificed by carbon dioxide. Blood samples for plasma lipid analysis were collected via the saphenous vein under light anaesthesia. Heart, liver, kidney, testis, perirenal fat pad, and epididymal fat pad were collected for relative organ calculation and stored at −80 °C until further analysis. The thoracic aorta was removed and preserved in DEPC-PBS solution at 4 °C for further atherosclerotic plaque analysis.

### 2.3. Animal Diet

Hamsters were fed either a control diet or one of the experimental diets enriched with Food Grade HCT extracts. A non-cholesterol diet (NCD) was formulated using the AIN-93 rodent diet base with slight modifications (g/kg diet): corn starch 508, casein 242, sucrose 119, lard 50, mineral mix 40, vitamin mix 20, DL-methionine 1, gelatin 20. A high-cholesterol diet (HCD) was formulated by incorporating 0.1% cholesterol into NCD in order to induce hypercholesterolemia in hamsters. Low-dosage and high-dosage diets were formulated by incorporating aqueous extracts and ethanolic extracts into HCD by 1% (L-, low dose) and 5% (H-, high dose), respectively (Table 1) [21,22].

### 2.4. Assessment of the Plasma Lipid Profile

Plasma levels of total cholesterol (TC), high-density lipoprotein cholesterol (HDL-C), and triacylglycerol (TG) were assessed using enzymatic kits from Stanbio Laboratories (Boerne, TX, USA) according to the manufacturer’s instructions. Non-HDL-C was calculated by subtracting HDL-C from TC.

### 2.5. Assessment of Aortic Fatty Streak Area

The percentage of fatty streak area in the endothelial layer of the thoracic aorta was quantified as previously described [23]. In brief, the thoracic aortas of the hamsters were carefully dissected, cleaned, and incised longitudinally. The aorta was subsequently stained with saturated oil red O (Cat No. 1320-06-5, Sigma Aldrich, St. Louis, MO, USA, dissolved in isopropanol) for 30 min. Afterwards, the aortas were then rinsed three times with isopropanol and distilled water. The fatty streaks were scanned and ImageJ software (FIJI v.1.0, National Institutes of Health, Bethesda, MD, USA) was used for quantification.

### 2.6. Assessment of Gut Microbial Community Composition

16S ribosomal RNA gene sequencing was performed following Liu’s method [24]. Briefly, fresh fecal samples were collected, and DNA was extracted using the E.N.Z.A.^®^ Stool DNA Isolation Kit (Omega Bio-Tek, Norcross, GA, USA). The barcoded universal bacterial primers 338F (ACTCCTACGGGAGGCAGCAG) and 806R (GGACTACHVGGGTWTCTAAT) were designed to target the variable 3–4 regions (V3–V4) of the 16S rRNA gene to amplify DNA. All samples were then sequenced using an Illumina MiSeq platform (Illumina, San Diego, CA, USA). QIIME 2 (version 2022.02) [25] was used to conduct the 16S rRNA gene sequence analysis. The demultiplexing sequences were reduced via the demux plugin. Subsequently, reads shorter than 270 and 240 nts in forward and reverse reads respectively, were denoised using dada2 [26] workflow to minimize sequencing error and eliminate chimeric reads. The filtered sequences were taxonomically classified using the Greengenes [27], 16S rRNA gene reference database using a pre-trained Naïve Bayes classifier [28] plugin by default parameters. The ASV table contained a range of 25,143 to 46,268 reads and was rarefied to 25,000 for each sample. The microeco package was used to calculate microbial community further, including α-diversity and β-diversity [28]. α-diversity was assessed by the Shannon and Simpson index, while β-diversity was calculated by Bray-Curtis distance [29], and the principal coordinates analysis was applied for ordination analysis. The pairwise Mann-Whitney U test was used for differential analysis, with a significance threshold of *p* < 0.05 for all comparisons unless otherwise stated. Functional capabilities of mucosal-associated microbiome were predicted using PICRUSt2 [30].

### 2.7. Statistical Analysis

All data were presented as means ± standard error of the mean (SEM). Data were analyzed using one-way ANOVA with *post hoc* Tukey’s analysis. Statistical analyses were conducted using GraphPad Prism software (Version 9.0, San Diego, CA, USA) to analyze differences among the groups. Statistical significance was defined as *p* < 0.05. The exact *p* value has been reported in the figures.

## 3. Results

### 3.1. Houttuynia cordata Thunb. (HCT) Extracts Do Not Alter Body Weight or Relative Organ Weights in Hamsters Fed with a High-Cholesterol Diet

First, we developed a hypercholesterolemic hamster model to elucidate the effects of HCT extracts on vascular health, cholesterol metabolism, and gut microbiota (Figure 1A). Hamsters were fed a high-cholesterol diet for six weeks to establish hypercholesterolemia. HCT extracts were mixed with the high-cholesterol diet and concurrently given to hamsters in this study. The efficacy of aqueous and ethanolic extracts of HCT was also evaluated and compared. Results revealed that no significant changes were detected in body weight across the six groups of hamsters throughout the experiment (Table 2). The relative liver weight was significantly higher in hamsters fed a high-cholesterol diet (HCD) compared to those fed with a non-cholesterol diet (NCD) (Table 2). It is worth noting that dietary supplementation with HCT extracts did not mitigate the increased liver weight in hypercholesterolemic hamsters (Table 2). In addition, there were no significant differences in the relative weights of the kidney, testis, or epididymal fat among the different groups (Table 2). These above findings demonstrate that both aqueous and ethanolic extracts of HCT do not change body weight or tissue weight in hamsters fed with a high-cholesterol diet.

### 3.2. Houttuynia cordata Thunb. (HCT) Extracts Reduce Atherosclerotic Plaques and Total Plasma Cholesterol Levels in Hypercholesterolemic Hamsters

Increasing total plasma cholesterol levels greatly promotes the progression of atherosclerosis, leading to vascular deaths. Hence, we next tried to determine whether HCT extracts alleviate atherosclerosis induced by hypercholesterolemia in the following studies. Aortas were isolated and dissected to remove perivascular adipose tissues. Then dissected aortas were stained with Oil Red O to characterize areas of atherosclerotic plaques in hamsters in different groups. The Oil red O results revealed that the areas of atherosclerotic plaques increased in the aortas derived from the HCD-fed hamsters. Impressively, both aqueous and ethanolic extracts of HCT dramatically decreased the atherosclerosis plaque areas in HCD-fed hamsters (Figure 1B,C), exhibiting the protective effects of HCT against hypercholesteremia-induced atherosclerosis. This is the first evidence demonstrating the alleviative effects of HCT on atherosclerosis, highlighting the potential of HCT as an agent and dietary supplement to improve vascular health upon stimulation of hyperchloremia.

Given that increasing total cholesterol is a notable contributing factor to atherosclerosis through thickening the arterial wall, we next evaluated the total cholesterol levels in different treatment groups. The plasma lipid profiles were similar among these six treatment groups at week 0. However, at the end of the six-week treatment period, continued consumption of HCD effectively triggered hypercholesterolemia in hamsters, exhibiting a significant increase in total cholesterol (TC) and non-HDL cholesterol by 27% and 54% in the plasma, respectively, in comparison to those fed with NCD. Encouragingly, the addition of L-HCEE and H-HCEE reduced plasma TC in hamsters fed with HCD, compared to the non-treated HCD group (Table 3 and Figure 1D). However, neither L-HCEE nor H-HCEE decreased the non-HDL cholesterol in HCD-treated hamsters (Table 3). In addition, HCAE treatment did not lower the plasma total cholesterol level or non-HDL cholesterol level in hamsters upon stimulation of a high-cholesterol diet (Table 3). These above findings demonstrate that only ethanolic extracts of HCT exhibit effects on lowering total cholesterol in hamsters. These findings also indicate that ethanolic extracts of HCT embrace better therapeutic effects on hypercholesterolemia and atherosclerosis compared to aqueous extracts of HCT.

### 3.3. Houttuynia cordata Thunb. (HCT) Modulates Gut Microbial Community Composition in Hypercholesterolemic Hamsters

Providing that gut microbiota plays an essential role in mitigating atherosclerosis and hypercholesterolemia. We next assessed whether HCT treatment changes the community composition of gut microbiota in hypercholesterolemic hamsters. Fresh feces were collected, and 16S rRNA sequencing was employed. Gut microbiota composition was compared among different treatment groups. The impact of HCT extracts on the gut microbial community was determined by comparing the differences in gut microbiota composition among normal cholesterol diet-treated hamsters, non-treated hypercholesterolemia hamsters, and HCT-treated hypercholesterolemia hamsters. The bacterial richness and diversity were assessed using the Shannon and Simpson indices (Appendix A). A Venn diagram was utilized to analyze the shared microbial richness across the six groups (Appendix A).

Results showed that six groups shared a total of 164 ASVs, with the NCD group displaying the highest number of ASVs. Principal Coordinate Analysis demonstrated a distinct clustering of bacterial compositions. After six weeks of high-cholesterol dietary intervention, the HCD group exhibited a different clustering pattern compared to the NCD group. However, the groups treated with HCT extracts, particularly the HCEE groups, showed a shift in microbiota composition towards that of the NCD group (Appendix A), indicating that HCT extracts may reverse the gut microbiota disorders caused by high cholesterol diet. Thus, these above findings demonstrate that HCT extracts alter the overall structure of the gut microbial community composition, illustrating that HCT extracts improve gut microbial community composition upon consumption of the high-cholesterol diet.

To capture the detailed changes in gut microbial community composition, the relative abundances of various gut microbiota were examined in different treatment groups at both the phylum and family levels (Figure 2). Impressively, results showed that HCT supplementation favourably modulates the composition of gut microbiota, which is in line with our previous notion. At the phylum level, the gut microbiota in hamsters was predominantly composed of *Firmicutes* and *Bacteroidetes* (Figure 2A). Strikingly, HCEE treatment remarkably enriched the population of *Bacteroidetes* while decreasing *Firmicutes*, resulting in a lower F/B ratio (Figure 2B,C). At the family level, the most prevalent families of gut microbiota were *Erysipelotrichaceae*, *Bacteroidales S24-7 group*, *Ruminococcaceae*, *Lachnospiraceae*, and *Porphyromonadaceae* in hamsters fed with the high-cholesterol diet (Figure 2D). Remarkably, *Porphyromonadaceae* levels increased in hamsters after receiving treatments of both aqueous and ethanolic extracts of HCT (Figure 2E). Besides, there was a trend of increasing *S24-7* abundance in both HCEE and HCAE treatment groups (Figure 2F). It has been reported that certain bacterial genera promote the proliferation of bacteria that produce SCFAs. Furthermore, previous studies have highlighted that significant changes in gut microbiota composition can lead to the reduction of plasma lipid levels [31]. Interestingly, at the genus level, HCT treatment significantly increased ASV_27, ASV_73, ASV_46, and ASV_76, compared to the HCD group (Figure 3). These above genera, which are increased by HCT treatment, all belong to the *S24-7* groups. Previous studies have revealed the tight correlation between increasing *S24-7* groups and improving vascular health. These findings indicate that HCT extract improves vascular health via increasing the beneficial bacterium.

### 3.4. Association between HCT-Mediated Shifts in Microbiota at the Genus Level and Its Effects on Lowering Plasma Lipid Levels

Given the pivotal role of gut microbiota in regulating cholesterol metabolism, we next characterize the correlation between HCT-modulated gut microbiota and its beneficial effects on lowering total plasma cholesterol levels. These results will pave the way for further exploration and development of HCT as a novel agent or dietary supplement for the treatment of hypercholesteremia and atherosclerosis. This study also offers valuable insights into the pharmacological effects of HCT in protecting against injuries induced by hypercholesteremia.

Therefore, Spearman’s correlation analysis was employed to explore the relationship between HCT-modulated shifts in intestinal microbiota at the genus level and its effects on reducing plasma lipid levels. By conducting Spearman’s analysis, the species interaction networks are constructed using visualization software that demonstrates species interrelationships. This analysis allows the search for mutually antagonistic or synergistic species within the microbial community in an environmental sample, which may guide treatment. Results demonstrated that the HCT-mediated reduction of plasma total cholesterol levels was positively correlated with the modulation of HCT in ASV_35, ASV_197, and ASV_260 (Figure 4A,B), indicating that HCT lowers total cholesterol levels via regulating gut microbial community composition.

## 4. Discussion

This present study characterizes the beneficial effects of *Houttuynia cordata* Thunb. extracts on protecting against hypercholesterolemia and atherosclerosis in hamsters fed with the high-cholesterol diet. Encouragingly, HCEE supplementation decreases total cholesterol levels and aortic plaque areas in hypercholesteremic hamsters. Furthermore, HCEE treatment favourably reverses gut microbial dysbiosis upon stimulation of hypercholesterolemia.

This study represents the first investigation into the beneficial effects of *Houttuynia cordata* Thunb. extracts on vascular health using a hypercholesterolemic hamster model. Moreover, this study demonstrated that the ethanolic extracts of HCT are more effective in reducing cholesterol levels compared to aqueous extract. By utilizing a human-like hypercholesterolemic model, this study provides valuable insights into the potential therapeutic benefits of HCT extracts. Results obtained from experiments using the hypercholesterolemic hamster model are more translatable due to their similar cholesterol metabolic mechanism to human hypercholesterolemia. The hamster model allows for the observation of the change in the aortic plaque area, which evaluates the effects of HCT extracts on vascular health. Besides, this model is able to replicate key features of hypercholesterolemic vascular dysfunction, such as plaque formation, which represents an ideal system for assessing the efficacy of HCT extract [32].

This study did not provide statistical analysis on the dose-dependent effects or compare the effects of different preparations of HCT, which is one of the limitations of this study. Although this study has fed hypercholesterolemic hamsters with different doses (low and high) and different preparations (aqueous and ethanolic) of HCT, the collected data is not enough to conduct two-way ANOVA to determine the dose-dependent effects of HCT. Therefore, further investigation can be conducted to explore the dose-dependent effects of HCT on reducing cholesterol and modulating gut microbiota. Besides, further research can also be conducted to determine which kinds of extraction of HCT, such as aqueous and ethanolic, exhibit better effects on decreasing cholesterol levels and regulating gut microbiota.

Furthermore, we recommend incorporating more refined methodologies to control and monitor the intake of HCT extract, such as oral gavage. Oral gavage allows for precise and consistent administration of hypercholesterolemic agents, thereby reducing variability in treatment dosage and ensuring more accurate assessments of their impact on vascular health [33]. This approach will enhance the reliability of the results and enable a more detailed exploration of the effects of HCT extract on various phenotypes associated with hypercholesterolemia. Oral gavage can also facilitate the investigation of dose-response relationships and time-dependent effects of hypercholesterolemic agents on vascular dysfunction [34]. By implementing this method, future studies could more effectively dissect the intricate mechanisms underlying lipid-induced vascular changes and evaluate the efficacy of therapeutic interventions.

The relationship between total cholesterol (TC) and the accumulation of atherosclerotic plaques is a critical focus of cardiovascular research [35]. Elevated TC is a well-established risk factor for the development and progression of atherosclerosis, which is characterized by the accumulation of lipid-laden plaques in arterial walls [36]. In our study, we observed a correlation between HCT-mediated decreased TC levels and reduced plaque formation, which is consistent with existing literature that supports the role of lipid management in mitigating atherosclerotic diseases [37]. Lower TC levels are often associated with a decrease in plaque area and improved vascular health, reinforcing the importance of lipid-lowering interventions in cardiovascular disease management [38].

The protective effects observed in this study suggest that HCT extracts may offer a new avenue for managing hypercholesterolemia and its associated vascular complications by decreasing the plasma total cholesterol. Despite these promising initial findings, it is important to acknowledge that our study did not comprehensively evaluate all the phenotypes related to hypercholesterolemia-induced atherosclerosis. Hypercholesterolemia impairs vascular function through various mechanisms, including endothelial dysfunction, oxidative stress, and vascular remodeling [39,40]. The potential effects of HCT extracts on reducing oxidative stress, enhancing endothelial function, or modulating lipid metabolism-deserve remain further investigated. Understanding these mechanisms could lead to the further development of effective therapeutic strategies for hypercholesterolemia-induced vascular diseases.

In addition, this study did not fully elucidate the mechanism by which HCT extracts lower cholesterol and modulate gut microbiota. The further elucidation of the specific molecular mechanisms through which HCT extracts exert their protective effects on atherosclerosis and gut microbiota disorder will enhance our understanding of the therapeutic potential of HCT on vasculopathy and metabolic diseases. The potential mechanism by which HCT extracts reduce cholesterol levels and atherosclerotic plaques could be its modulation of glycan metabolism, fructose and mannose metabolism, and fatty acid biosynthesis through remodeling microbiota composition.

Recently, more and more research has started to focus on the pharmacological effects of bioactive compounds present in plants and Chinese herbs [41]. Previous studies have demonstrated that HCT exhibits various biological activities, including anti-inflammatory, antiviral, antibacterial, and antioxidant. These activities are attributed to various bioactive compounds, which can be extracted using different solvents. Aqueous is ideal for extracting polar compounds such as polysaccharides, glycosides, and certain alkaloids. These compounds might have anti-inflammatory or antimicrobial properties [12,13,14,15]. Typically, people consume HCT directly and by boiling it in water. Ethanol is a polar organic solvent that can extract both polar and non-polar compounds, particularly flavonoids, terpenoids, and other phenolic compounds, which exhibit activities including antioxidant, anti-inflammatory, and antiviral. Flavonoids such as quercetin and rutin are abundant in HCT ethanolic extract but can be dissolved in aqueous extract of HCT as well.

Flavonoids such as quercetin and rutin are abundant in HCT ethanolic extract. Flavonoids are known to regulate lipid metabolism by inhibiting cholesterol absorption [42]. Polyphenols in HCT ethanolic extracts might play a role in modulating lipid profiles. Polyphenols can influence lipid metabolism by inhibiting lipid peroxidation. The oxidation of LDL cholesterol is a key step in the development of atherosclerosis [43]. Some studies have proved ethanolic extracts lower blood lipids in vivo, but whether they can decrease cholesterol levels and remodel the gut microbiota profile is still unknown [19]. Comparing the effects of ethanolic and aqueous extracts provides valuable insights into whether the major bioactive compounds in HCT are polar or non-polar, facilitating further investigation into the mechanism by which HCT reduces total cholesterol levels and modulates gut microbiota.

However, the specific bioactive compounds in HCT extracts that regulate microbiota composition and reduce cholesterol levels remain to be investigated. Further Investigations can focus on the dosage, long-term effects, and potential interactions of HCT with other cardiovascular therapies using other experimental models, such as dogs and monkeys, which is essential for translating these findings into clinical practice. Moreover, the scope of this study is limited to assessing vascular health without a comprehensive evaluation of the potential systemic effects of HCT extract. Future studies can address these limitations by incorporating diverse models and broader assessments of the impact of HCT on overall health.

Our study has identified several candidate genera modulated by HCT extracts, which are associated with the modulation of gut microbiota profiles and cholesterol levels. These findings indicate that specific gut microbiota profiles are linked to HCT extract-mediated cholesterol-lowering effects, which provides a new avenue for therapeutic interventions aimed at managing hypercholesterolemia [44]. However, further research is suggested to fully elucidate the role of HCT-modulated genera and the specific contributions to its cholesterol regulation. Following the determination of the candidate genera, a crucial next step involves evaluating additional parameters related to their functional contributions. This includes investigating how these genera influence cholesterol metabolism, bile acid synthesis, and overall gut health. Parameters such as the production of short-chain fatty acids (SCFAs), modulation of systemic inflammation, and interactions with host lipid metabolism pathways should be assessed. Detailed metabolic profiling and functional assays will provide deeper insights into how these microbiota components contribute to cholesterol-lowering effects.

Fecal microbiota transplantation (FMT) represents a powerful approach to verifying the specific role of identified genera in cholesterol modulation [45]. FMT involves transferring fecal microbiota from a donor with a well-characterized microbiota profile into a recipient, thereby introducing the specific microbiota composition associated with cholesterol-lowering effects. This method can help establish a causal relationship between the candidate genera and their impact on cholesterol levels. Building upon our current findings, FMT is encouraged to be conducted to further evaluate the relationship between gut microbiota and HCT-mediated vascular improvements, providing valuable insights into the therapeutic effects of HCT on hypercholesterolemia-induced vasculopathy.

Thus, this study represents the first exploration of the vascular protective effects of HCT extract in the hypercholesterolemic hamster model, highlighting the potential of HCT as a therapeutic agent for the treatment of hypercholesterolemia-related vascular diseases. Building upon our above findings, this research lays the foundation for further investigations into the clinical applicability of HCT. Besides, these findings underscore the importance of further studies elucidating the mechanisms of HCT extracts in improving vascular health and modulating gut microbiota, offering valuable insights into the development of novel interventions for the treatment of hypercholesterolemia and atherosclerosis.

Although our study has identified candidate genera that modulate gut microbiota profiles total cholesterol levels and alleviate atherosclerosis, further research is encouraged to delve into their correlation. A comprehensive evaluation of the vascular protective effects of HCT extract and fecal microbiota transplantation will offer promising insights into the specific roles of HCT in preventing atherosclerosis and hypercholesterolemia. These findings can not only improve the understanding of the role of gut microbiota in improving cardiovascular health but also explore novel microbiota-based therapeutic strategies for managing hypercholesterolemia-related diseases.

## 5. Conclusions

This study demonstrates, for the first time, the beneficial effects of *Houttuynia cordata* Thunb. extracts on improving vascular health upon stimulation of hypercholesterolemia. High dosage intake of ethanolic extracts of *Houttuynia cordata* Thunb. exhibits a more significant reduction of total plasma cholesterol levels compared with aqueous extracts. Oral treatment of *Houttuynia cordata* Thunb. extracts favourably modulate the gut microbiota in hypercholesterolemic hamsters by increasing *Bacteroidetes*, decreasing *Firmicutes*, and enhancing the growth of SCFA-producing bacteria. This study demonstrates the positive correlation between HCT-mediated reduction of the total plasma cholesterol levels and HCT-modulated gut microbial community composition in hamsters fed with the high-cholesterol diet.

## Figures and Tables

**Figure 1 nutrients-16-03290-f001:**
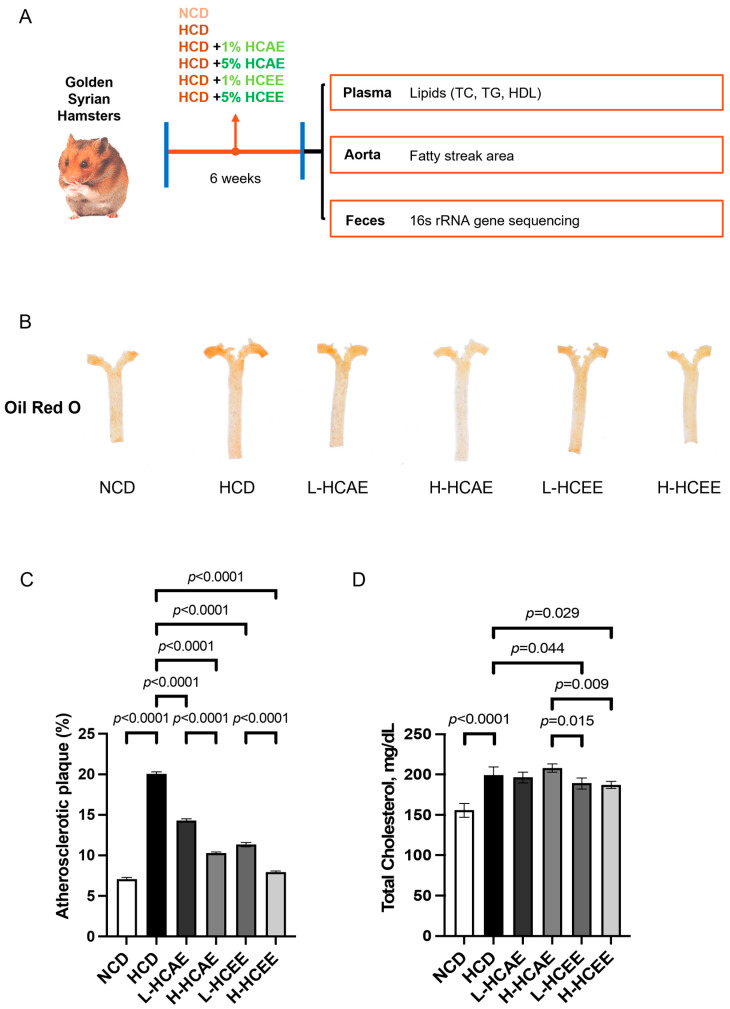
The beneficial effects of *Houttuynia cordata* extract on vascular health in high cholesterol diet-induced atherosclerosis hamster model. (**A**) Experimental design of animal study. (**B**,**C**) Plaques increase in aortas derived from the HCD-fed hamsters while both aqueous and ethanolic extracts of *Houttuynia cordata* treatment significantly decrease the atherosclerosis plaques area in HCD-fed hamsters. (**D**) Total plasma cholesterol levels are significantly decreased in ethanolic extracts of *Houttuynia cordata*-treated HCD-fed hamsters. Data represent mean ± SEM of 7–9 hamsters and analyzed by one-way ANOVA with *post hoc* Tukey’s analysis. NCD, normal cholesterol diet; HCD, high cholesterol diet; HCAE, *Houttuynia cordata* aqueous extract; HCEE, *Houttuynia cordata* ethanolic extract; L-HCAE, high cholesterol diet containing 1% *Houttuynia cordata* aqueous extract; H-HCAE, high cholesterol diet containing 5% *Houttuynia cordata* aqueous extract; L-HCEE, high cholesterol diet containing 1% *Houttuynia cordata* ethanolic extract; H-HCEE, high cholesterol diet containing 5% *Houttuynia cordata* ethanolic extract.

**Figure 2 nutrients-16-03290-f002:**
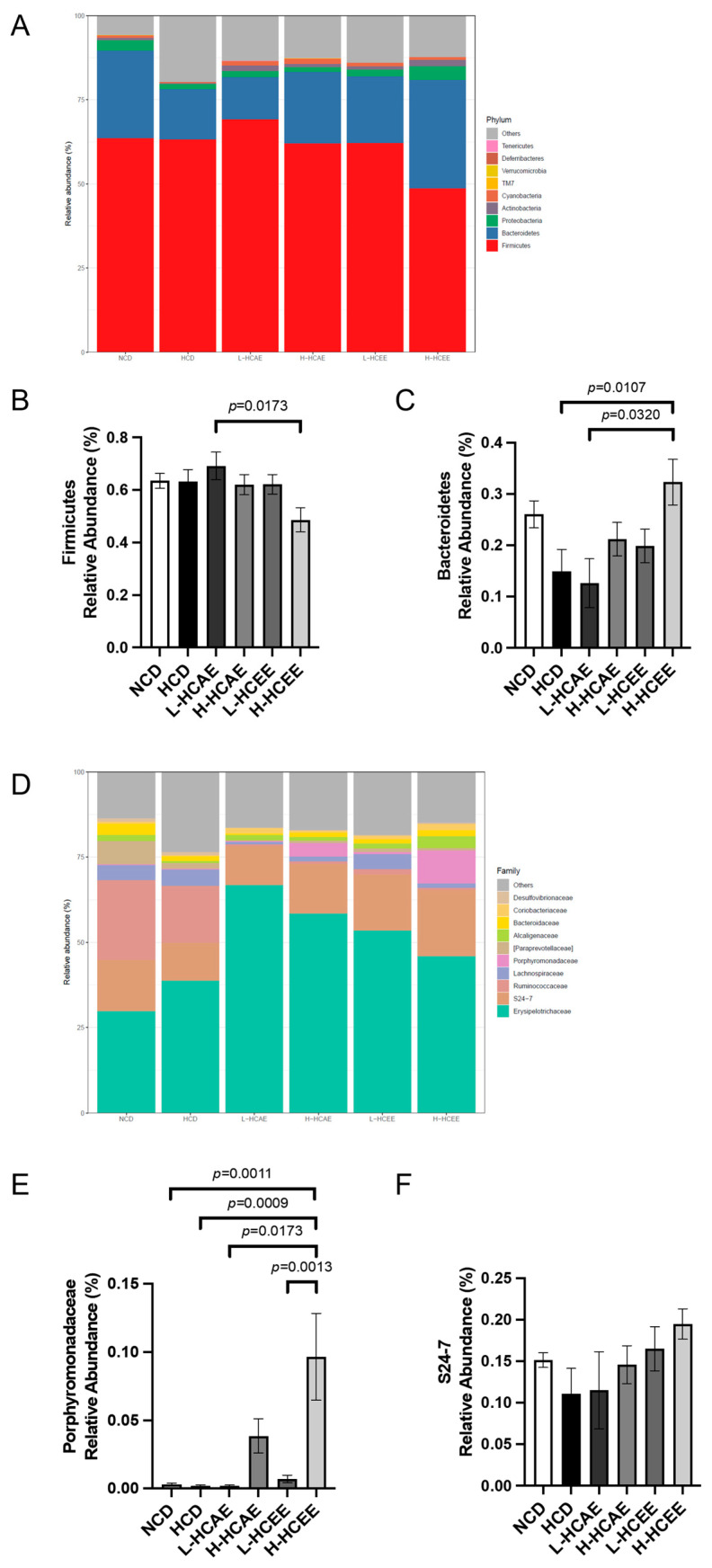
*Houttuynia cordata* extracts modulate gut microbial composition in hamsters fed with HCD. Hamsters are fed with NCD, HCD, L-HCAE, H-HACE, L-HCEE, and H-HCEE for 6 weeks, respectively. (**A**) Relative abundance of the top dominant gut microbiota of each group on the phylum level. (**B**,**C**) Relative abundance of *Firmicutes* decreases in *Houttuynia cordata* ethanolic extracts-treated hamsters (**B**) while *Bacteroidetes* increases after 5% *Houttuynia cordata* ethanolic extracts treatment (**C**). (**D**) Relative abundance of the top dominant gut microbiota of each hamster on family level. (**E**) Relative abundance of *Porphyromonadaceae* increases in both 5% *Houttuynia cordata* aqueous and ethanolic extracts-treated hamsters. (**F**) Relative abundance of *S24-7* increases in both *Houttuynia cordata* aqueous and ethanolic extracts-treated hamsters. Data represent mean ± SEM of 7–9 hamsters and analyzed by one-way ANOVA with *post hoc* Tukey’s analysis. NCD, normal cholesterol diet; HCD, high cholesterol diet; HCAE, *Houttuynia cordata* aqueous extract; HCEE, *Houttuynia cordata* ethanolic extract; L-HCAE, high cholesterol diet containing 1% *Houttuynia cordata* aqueous extract; H-HCAE, high cholesterol diet containing 5% *Houttuynia cordata* aqueous extract; L-HCEE, high cholesterol diet containing 1% *Houttuynia cordata* ethanolic extract; H-HCEE, high cholesterol diet containing 5% *Houttuynia cordata* ethanolic extract.

**Figure 3 nutrients-16-03290-f003:**
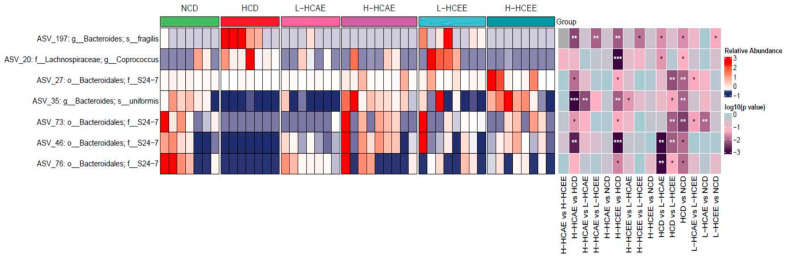
Heatmap of selected gut microbiota at the genus level based on relative abundance. Hamsters are fed with NCD, HCD, L-HCAE, H-HACE, L-HCEE, and H-HCEE for 6 weeks, respectively. The blue to red in the picture represents the relative abundance at the genus level that changes from lower to greater. Data represent mean ± SEM of 7–9 hamsters. Asterisks “*”, “**”, and “***” indicate the different levels of associations significant at *p* < 0.05, *p* < 0.01, and *p* < 0.001, respectively, by one-way ANOVA with *post hoc* Tukey’s analysis. The group comparison information of the genera is displayed on the right. NCD, normal cholesterol diet; HCD, high cholesterol diet; HCAE, *Houttuynia Cordata* aqueous extract; HCEE, *Houttuynia cordata* ethanolic extract; L-HCAE, high cholesterol diet containing 1% *Houttuynia cordata* aqueous extract; H-HCAE, high cholesterol diet containing 5% *Houttuynia cordata* aqueous extract; L-HCEE, high cholesterol diet containing 1% *Houttuynia cordata* ethanolic extract; H-HCEE, high cholesterol diet containing 5% *Houttuynia cordata* ethanolic extract.

**Figure 4 nutrients-16-03290-f004:**
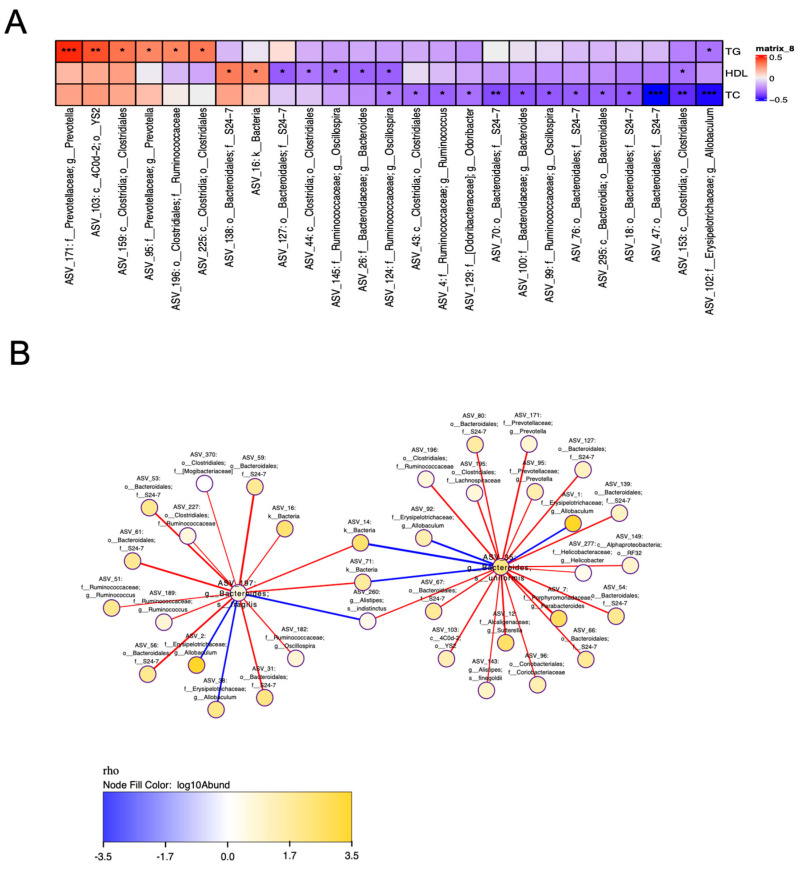
Spearman’s correlation analysis. Correlation analysis between key genera and plasma lipid profile (**A**) and correlation analysis between key genera (**B**). Hamsters are fed with NCD, HCD, L-HCAE, H-HACE, L-HCEE, and H-HCEE for 6 weeks, respectively. Colours of squares indicate the R-value of Spearman’s correlation. Data are represented as mean ± SEM of 7–9 hamsters. Asterisks “*”, “**”, and “***” indicate the different levels of associations significant at *p* < 0.05, *p* < 0.01, and *p* < 0.001, respectively, by one-way ANOVA with *post hoc* Tukey’s analysis. NCD, normal cholesterol diet; HCD, high cholesterol diet; HCAE, *Houttuynia cordata* aqueous extract; HCEE, *Houttuynia cordata* ethanolic extract; L-HCAE, high cholesterol diet containing 1% *Houttuynia cordata* aqueous extract; H-HCAE, high cholesterol diet containing 5% *Houttuynia cordata* aqueous extract; L-HCEE, high cholesterol diet containing 1% *Houttuynia cordata* ethanolic extract; H-HCEE, high cholesterol diet containing 5% *Houttuynia cordata* ethanolic extract.

**Table 1 nutrients-16-03290-t001:** Composition of experimental diets (g/kg).

Ingredients (g/kg Diet)	NCD	HCD	L-HCAE	H-HCAE	L-HCEE	H-HCEE
Corn starch	508	508	508	508	508	508
Casein	242	242	242	242	242	242
Sucrose	119	119	119	119	119	119
Lard	50	50	50	50	50	50
Mineral mixture	40	40	40	40	40	40
Vitamin mixture	20	20	20	20	20	20
Gelatin	20	20	20	20	20	20
DL-Methionine	1	1	1	1	1	1
Cholesterol		1	1	1	1	1
HCAE	-	-	10	50	-	-
HCEE					10	50

NCD: non-cholesterol diet; HCD: 0.1% cholesterol added into NCD diet; L-HCAE: 1% *Houttuynia cordata* aqueous extract (HCAE) added into HCD; H-HACE: 5% HCAE added into HCD; L-HCEE: 1% *Houttuynia cordata* ethanolic extract (HCEE) added into HCD; H-HCEE: 5% HCEE added into HCD.

**Table 2 nutrients-16-03290-t002:** Food intake, body weight, and the relative organ weights in hamsters.

	NCD	HCD	L-HCAE	H-HCAE	L-HCEE	H-HCEE	*p* Value
Daily food intake, g	8.51 ± 0.24 ^a^	7.88 ± 0.13 ^b^	8.42 ± 0.10 ^a^	8.53 ± 0.15 ^a^	8.19 ± 0.26 ^ab^	8.57 ± 0.02 ^a^	<0.01
Body weight, g							
Initial (Week 0)	111.30 ± 2.04	112.18 ± 3.13	115.00 ± 2.22	115.08 ± 2.01	111.00 ± 2.02	112.38 ± 2.11	0.52
Final (Week 6)	118.00 ± 2.38	115.04 ± 2.58	123.25 ± 3.31	125.08 ± 3.06	124.24 ± 3.00	124.12 ± 2.04	0.05
Body weight gain	6.70 ± 1.23	2.86 ± 1.77	8.25 ± 2.46	10.00 ± 2.31	13.24 ± 2.22	11.74 ± 0.54	0.05
Relative organ weight, % of body weight (Week 6)							
Liver	3.29 ± 0.10 ^b^	3.84 ± 0.12 ^a^	3.85 ± 0.07 ^a^	3.83 ± 0.06 ^a^	3.95 ± 0.11 ^a^	3.92 ± 0.05 ^a^	<0.001
Heart	0.45 ± 0.03 ^a^	0.40 ± 0.01 ^a^	0.37 ± 0.01 ^b^	0.41 ± 0.01 ^a^	0.41 ± 0.01 ^a^	0.38 ± 0.01 ^ab^	0.02
Kidney	0.74 ± 0.02	0.80 ± 0.01	0.72 ± 0.01	0.75 ± 0.01	0.74 ± 0.02	0.74 ± 0.04	0.08
Perirenal fat pad	1.09 ± 0.11 ^b^	1.08 ± 0.05 ^b^	1.24 ± 0.06 ^ab^	1.23 ± 0.09 ^ab^	1.12 ± 0.05 ^b^	1.44 ± 0.07 ^a^	<0.01
Testis	3.97 ± 0.33	3.84 ± 0.18	3.67 ± 0.18	3.75 ± 0.06	3.43 ± 0.16	3.65 ± 0.28	0.52
Epididymal fat pad	1.70 ± 0.09	1.69 ± 0.11	1.83 ± 0.07	1.81 ± 0.11	1.73 ± 0.09	1.85 ± 0.14	0.75

NCD: non-cholesterol diet; HCD: 0.1% cholesterol added into NCD diet; L-HCAE: 1% *Houttuynia cordata* aqueous extract (HCAE) added into HCD; H-HACE: 5% HCAE added into HCD; L-HCEE: 1% *Houttuynia cordata* ethanolic extract (HCEE) added into HCD; H-HCEE: 5% HCEE added into HCD. Data represent mean ± SEM of 7–9 hamsters. Means denoted with different superscript letters (a,b) differ significantly, *p* < 0.05 by one-way ANOVA with *post hoc* Tukey’s analysis.

**Table 3 nutrients-16-03290-t003:** Basic parameters including plasma lipid total cholesterol (TC), high-density lipoprotein cholesterol (HDL-C), triacylglycerol (TG), non-HDL cholesterol (non-HDL-C), and ratio of HDL-C/TC and non-HDL-C/HDL-C in different treatment groups.

	NCD	HCD	L-HCAE	H-HCAE	L-HCEE	H-HCEE	*p* Value
TC, mg/dL	156 ± 9 ^c^	199 ± 10 ^a^	196 ± 7 ^a^	208 ± 5 ^a^	189 ± 7 ^b^	187 ± 4 ^b^	<0.001
HDL-C, mg/dL	131 ± 4 ^c^	162 ± 7 ^a^	148 ± 6 ^ab^	162 ± 4 ^a^	136 ± 3 ^b^	144 ± 4 ^b^	<0.001
Non-HDL-C, mg/dL	24 ± 9 ^b^	37 ± 10 ^ab^	48 ± 5 ^ab^	46 ± 2 ^ab^	53 ± 4 ^a^	43 ± 5 ^ab^	0.03
HDL-C/TC	0.86 ± 0.04 ^b^	0.82 ± 0.04 ^ab^	0.76 ± 0.02 ^ab^	0.78 ± 0.01 ^ab^	0.72 ± 0.01 ^a^	0.77 ± 0.02 ^ab^	0.02
Non-HDL-C/HDL-C	0.19 ± 0.07 ^b^	0.23 ± 0.06 ^ab^	0.33 ± 0.04 ^ab^	0.28 ± 0.02 ^ab^	0.39 ± 0.02 ^a^	0.31 ± 0.04 ^ab^	0.04
TG, mg/dL	47 ± 5	50 ± 3	71 ± 7	68 ± 9	67 ± 8	57 ± 8	0.13

NCD: non-cholesterol diet; HCD: 0.1% cholesterol added into NCD diet; L-HCAE: 1% *Houttuynia cordata* aqueous extract (HCAE) added into HCD; H-HACE: 5% HCAE added into HCD; L-HCEE: 1% *Houttuynia cordata* ethanolic extract (HCEE) added into HCD; H-HCEE: 5% HCEE added into HCD. Data represent mean ± SEM of 7–9 hamsters. Means denoted with different superscript letters (a–c) differ significantly, *p* < 0.05 by one-way ANOVA with *post hoc* Tukey’s analysis.

## Data Availability

The original contributions presented in the study are included in the article, further inquiries can be directed to the corresponding author.

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
