# Peer review of "Houttuynia cordata Thunb. Extracts Alleviate Atherosclerosis and Modulate Gut Microbiota in Male Hypercholesterolemic Hamsters"

_nutrients, 2024, doi:10.3390/nu16193290_

Round 1

Reviewer 1 Report

Comments and Suggestions for Authors

Abstract - Line 31 – the effects are “greater” but not “better”.

Introduction - The paragraph describing HCT only provided a board overview. The rationale of focusing on the effects of HCT on hyperchloremia is not clearly specified.

Methods

1.     Line 100 & 101– past tense needs to be used.

2.     Table 1 - What was the reason for the significant difference in the diet composition?

3.     Was the animal pair-fed or ad libitum? How was the dosage of the HCT determined?

4.     What was the rationale for using both ethanolic and aqueous extract of HCT? If the active ingredients in different extracts account for different actions, please provide evidence to support the design.

Results – It is better to use the same pattern for the same group in the bar chart throughout the manuscript to keep consistent.

Discussion – Line 316-335 A further discussion based on existing evidence, even if it is not direct evidence on HCT, should be included to elucidate the potential mechanisms of the effect of HCT in managing hypercholesterolemia. The direction of future studies brought by the authors is too board and general but not specific enough. This is also related to the comments on the Introduction that the authors need to provide a stronger rationale for the study design.

Comments on the Quality of English Language

The English language is generally good. Minor style and spelling need to be double-checked before publication.

Reviewer 2 Report

Comments and Suggestions for Authors

Lin. et all demonstrates that Houttuynia cordata Thunb (HCT) extract significantly reduces cholesterol levels and atherosclerotic plaques in hypercholesterolemic hamsters. Both aqueous and ethanolic extracts of HCT positively modulate gut microbiota, promoting the growth of beneficial bacteria. These findings highlight the potential of HCT as a dietary supplement for maintaining cholesterol homeostasis and improving vascular health.

Although study looks good, author need to address a lot of concern to make this manuscript readable

·         Why do you treat HCT with two different types of solutions, aqueous and ethanolic?

·         Is there any literature that shows the facilitation of short-chain fatty acids contributes to the reduction of plasma lipid levels?

·         In Figure 1B, clarify the legend. Is it ORO staining?

·         Why did the total final body weight not increase (and even seem to decrease) in the HCD group? Additionally, why did co-treatment with both ethanolic and aqueous HCT not result in a decrease in the final body weight of the hamsters?

·         Make Figure 2 larger as it is not clearly visible.

·         Every figure in the document is unclear and needs improvement.

Reviewer 3 Report

Comments and Suggestions for Authors

Thank you for submitting the manuscript "Houttuynia cordata Thunb. Extracts Alleviate Atherosclerosis and Modulate Gut Microbiota in Hypercholesterolemic Hamsters" to Nutrients. Although the topic of the manuscript is within the scope of Nutrients, the work lacks consensus on several points and therefore it is not possible to perform a more in-depth evaluation of the results obtained with the research. Here are some points that need to be checked:

- the sex of the hamsters used in the study should be added to the title.

- Line #38: is this statement based on the change in abundance of microorganisms in the microbiota or were fatty acids also evaluated? If yes, consider modifying line #29. If not, consider changing the statement.

- Lines #39-43: consider rewriting for clarification.

- Line #45: food or phytotherapeutic or nutraceutical or supplement? - Line#84: in fact, what was investigated was not the protective effect, since hypercholesterolemia was apparently already present. Consider modifying.

- Lines#92-82: Too hopeful based on the methodology proposed in the work.

- Line#107: more information about the extract should be added, such as a basic chemical composition, quantity of phenolic compounds, etc.

- Line#108: consider including which part of the plant was used in the study.

- Line#111: it is necessary to add how long the animals were fed the hypercholesterolemic diet for the development of the disease before starting the extract. If this was not done, it is necessary to review other points of the work where it was indicated in the text that the animals already had the disease and that the extract helped in the treatment. If the diet was started together with the treatment with the extract, the statements made throughout the text should be changed. - Line #119: if only one stool sample was collected in week 6, how did the authors state in other parts of the manuscript that the microbiota was modified without using a comparison with the beginning of the study? Review the entire manuscript for this issue.

- Table 2: the most important result here regarding weight is weight gain and food intake. They need to be included in the table and discussed in the text.

- The tables appear to be in figure format. Consider inserting them as tables in Microsoft Word and formatting them according to the manuscript standard.

- What statistical analysis does the p value in tables 2 and 3 refer to? Normality test or analysis of mean test between samples? Clarify. If it refers to the mean test of the samples, letters need to be added to indicate where this difference exists.

- It is impossible to see figures 2, 3 or 4. The resolution is very poor and so is the size.

- Consider including the limitations of the work and also what would be the "next steps" of the research or next steps that could clarify weak points of this work.

Comments on the Quality of English Language

 Minor editing of English language required

Reviewer 4 Report

Comments and Suggestions for Authors

Houttuynia cordata Thunb (fishy Chinese herb) is well-known in China and is a food medicinal herb exhibiting various pharmacological properties. The alleviative effects of Houttuynia cordata Thunb on hyperchloremia and atherosclerosis remain largely investigated. The manuscript (nutrients-3158776-peer-review-v1) submitted by Yuhong Lin et al. describes the effects of HCT extracts on vascular health and gut microbiota in golden Syrian hamsters with hypercholesterolemia: Houttuynia cordata Thunb. Extracts Alleviate Atherosclerosis and Modulate Gut Microbiota in Hypercholesterolemic Hamsters. The conclusions indicate that the potential of HCT as a dietary supplement to alleviate atherosclerosis, lower plasma cholesterol and modulate gut microbiota. The content of this MS is abundant. However, this MS needs revision before publication. See my comments below to improve this manuscript.

1. Please provide much more information about HCT in the manuscript, such as the main compounds.

2. The quality of Figures should be improved, I cant see the picture content clearly.

3. The decimal places need to be consistent in table 2. The representation of numbers should be scientific numeration.

4. Which compounds have an effect on gut microbiota? What is the concentration effect? What is the time effect?

Round 2

Reviewer 2 Report

Comments and Suggestions for Authors

The author answered my query

Author Response

Sincerely thanks for your comments.

Reviewer 3 Report

Comments and Suggestions for Authors

This reviewer thanks the authors for incorporating all of the suggestions made. The manuscript has greatly improved in quality and clarity and therefore, I believe it can be accepted for publication.

Author Response

Sincerely thanks to your affirmation.

Reviewer 4 Report

Comments and Suggestions for Authors

The authors show off their style. I would praise their work.

Author Response

Sincerely thanks to your praise.